# Potential Global Distribution of *Daktulosphaira vitifoliae* under Climate Change Based on MaxEnt

**DOI:** 10.3390/insects12040347

**Published:** 2021-04-13

**Authors:** Wei Ji, Gary Gao, Jiufeng Wei

**Affiliations:** 1College of Horticulture, Shanxi Agricultural University, Taigu 030801, China; jiweiputao@163.com; 2Department of Horticulture and Crop Science, The Ohio State University, 2001 Fyffe Court, Columbus, OH 43210, USA; Gao.2@osu.edu; 3College of Plant Protection, Shanxi Agricultural University, Taigu 030801, China

**Keywords:** grape phylloxera, MaxEnt, habitat suitability simulation, climate change

## Abstract

**Simple Summary:**

*Daktulosphaira vitifoliae* (Fitch) or grape phylloxera is a small, invasive, and sap-sucking insect widely distribution in most viticultural areas of the world. In the current study, the potential distribution ranges of the leaf-feeding population under current and future environmental conditions were simulated by MaxEnt software. The highly suitable ranges of *D. vitifoliae* mainly focus on Europe, East and North China, Japan, the Eastern USA, Uruguay, and the Southeast of South America under current climatic conditions. The highly suitable ranges were obviously increased under future climate conditions.

**Abstract:**

Grape phylloxera, *Daktulosphaira vitifoliae*, is a small, invasive, sap-sucking pest that is widely present in most viticulture regions all over the world. It is originally from North America and feeds on grapevine roots and leaves. In the current study, the potential distribution area of the leaf-feeding population was investigated with MaxEnt based on population occurrence data under different environmental variables. Results suggested that under current climatic conditions, Europe, East and North China, Japan, the Eastern USA, Uruguay, and the Southeast of South America are highly suitable areas for the occurrence of phylloxera leaf populations. The results showed that isothermality and precipitation of coldest quarter were major factors which contribute more than 60% of the model under current climate conditions. Our results provide important information for governmental decision makers and famers to develop control and management strategies against *D. vitifoliae*, and can also be used as a reference for studies on other invasive pest.

## 1. Introduction

*Daktulosphaira vitifoliae* (Fitch) or grape phylloxera is a small, invasive, and sap-sucking insect widely distributed in most viticultural regions of the world [1,2]. Grape phylloxera has two different types, the radicicoles and the gallicoles, which affect roots and leaves, respectively [3]. Both types of root damage facilitate soil-borne pathogens to enter into the root cortex, resulting in growth reductions, yield loss or even death [4,5].

Grape phylloxera usually spreads between vineyard through human pathways, including footwear, clothing, machinery, and plant material [6]. Moreover, first instars or winged of this species can migrate by wind [3]. This species is native to North America and now grows in most viticultural countries worldwide [7]. Over the past 150 years, this pest has spread quickly to most wine grape growing regions of the word, including South America, New Zealand, Australia, South Africa, and China [7]. Therefore, there is an urgent need to improve control and management methods for *D. vitifoliae* due to its continuous spread on a global scale. 

Grapevine rootstocks are recognized as having intrinsic resistance and tolerance mechanisms to control grape phylloxera. Most of the rootstocks used in modern viticulture are ranked tolerant against *D. vitifoliae*; thus, they can provide habitat and food for pest populations without causing damage to the host [8]. However, rootstocks are the only viable long-term solution for the management of grape phylloxera; however, in some countries or regions, rootstocks are still an uneconomic option. Moreover, rootstocks breakdown or failure in some regions is due to the presence of more aggressive phylloxera biotypes [8,9]. Therefore, an alternative method might be to provide useful information for preventing the spread of the pest into regions or countries where grape phylloxera has not yet been recorded. 

In addition, climate changes can influence the occurrence of species by affecting the population growth, reproductive phenology [10,11], and transmission [12], thereby significantly affecting their distribution [13]. Identifying the potential distribution areas of this invasive pest under different climate conditions is an important step to formulate strategies that hinder and limit the expansion of grape phylloxera. Moreover, understanding the potential distribution area of the grape phylloxera under climate change will be necessary for scientists and farmers, as they may formulate future monitoring and management strategies [14]. There are some field studies that explore the distribution of *D. vitifoliae* by analysis of the soil properties [15], climate change [1,16], and management techniques based on geographical sites [17]. However, the potential distribution region of this pest is unclear. This may seriously affect the formulation of control strategies and recommended measures for this pest. In order to better prevent the spread of *D. vitifoliae*, a global potential distribution forecast is still needed. Species distribution models (SDMs) might help solve this problem. Over the past ten years, SDMs have been frequently employed as a powerful tool that use climatic and occurrence records to infer the potential geographic distribution [18,19]. 

In the present study, the occurrence data and the environmental variables were employed to model the potential global distribution under current and future climate conditions and identify the main climatic factors that form the current potential distribution pattern. In addition, this research will also provide theoretical references for formulating policies for pest management and control. 

## 2. Materials and Methods

### 2.1. Occurrence Sites

Information on occurrence sites of *D. vitifoliae* used in the current paper was compiled from the following sources: Global Biodiversity Information Facility (GBIF, https://www.gbif.org/, accessed on 3 November 2020), the Centre of Agriculture and Bioscience International (CABI, https://www.cabi.org/, accessed on 15 November 2020), and previously published literature (see Appendix A). Google Earth or literature was applied to assign geographical coordinates for the distribution sites which any records lacking geo-coordinates (just record administrative region). A total of 128 distribution sites were documented in the initial study. 

Sampling bias and spatial autocorrelation might affect the results of the model [20]. Completely eliminating the sample bias and spatial autocorrelation remains challenging; however, filtering occurrence can improve model performance [21]. Thus, a coarse resolution (5 km × 5 km) was created, and a single occurrence location was randomly kept from each cell with one or more occurrence locations in current study [22].

This work was implemented in ArcGIS 10.1 (ESRI, Redlands, CA, USA), and after filtering, 103 location remained. 

### 2.2. Environment Data 

Nineteen environmental variables with 2.5 arc min spatial resolution (~5 km) were obtained from WorldClim Global Climate Database (http://www.worldclim.org, accessed on 20 November 2020) in an initial study [23]. These climatic variables were derived from the monthly temperature and precipitation, seasonal variation, and climatic extreme indices covering a period of time from 1950 to 2000 [24]. Topographic variables represented by altitude (Alt) were derived from HYDRO1K (http://eros.usgs.gov, accessed on 20 November 2020). Environmental variables used in SDMs may directly or indirectly affect the performance of model [25]. 

Multicollinearity may lead to over-fitting and might reduce the precision of the model [25,26]. Therefore, to reduce the multicollinearity among the climate variables, Pearson’s correlation coefficient was employed in this study. When two environmental variables had a correlation coefficient (R ≥ |0.8|) (Appendix A), only one of the pair was the choice for the model based on percent deviances explained from a univariate generalized additive model with the predictor [27]. Pearson’s correlation was carried out in SPSS statistics software (IBM Corp., Armonk, NY, USA). 

As a result, seven climatic variables were used in the present study (Table 1): Bio2 (Mean diurnal range (Mean of monthly (max temp–min temp)), Bio3 (Isothermality (BIO2/BIO7) (*100)), Bio8 (Mean temperature of wettest quarter), Bio12 (Annual precipitation), Bio18 (Precipitation of warmest quarter), Bio19 (Precipitation of coldest quarter), and Alt (Altitude).

To predict the potential distribution of the invasive pest under future climate conditions on a global scale, the Hadley Global Environment Model 2-Atmosphere Ocean (HADGEM2-AO), Geophysical Fluid Dynamics Laboratory Coupled Model v3 (GFDL-CM3), and Model for Interdisciplinary Research on Climate (MIROC5) for 2050 (average for 2041–2060) and 2070 (average for 2061–2080) were defined from the Coupled Model Inter-comparison Project Phase 5 (CMIP5) of the fifth assessment of the Intergovernmental Panel on Climate Change (IPCC) in 2010 [28]. The fifth assessment report was published by the UN’s Intergovernmental Panel on Climate Change (IPCC) with four representative concentration pathways (RCP), i.e., RCP 2.6, RCP 4.5, RCP 6.0, and RCP8.5 [29]. These four pathways are defined by the possible greenhouse gas emission trajectories (ranging from lowest (2.6) to highest (8.5)), equivalent to the increments in global radiative forcing values in the year 2100 [30]. Extreme climatic variations may have greater impact on the potential distribution model; thus, four climate change scenario/year combinations were used in current research: RCP-2.6-2050, RCP 8.5-2050, RCP 2.6-2070, and RCP 8.5-2070. The current and future climatic scenarios were re-sampled to 2.5 arc-min for further modeling.

### 2.3. Modelling Approach

The MaxEnt modeling version 3.4.1 was employed in the current study [31]. The regularization multiplier (RM) and the feature combination (FC) may influence the predictive performance and accuracy of the model. To get the best model and control overparameterization, the R package “ENMeval” was employed [32]. The RM ranges from 0.5 to 4 in increments of 0.5, while seven FCs were used in current study: (1) Linear (L); (2) Linear (L) and Quadratic (Q); (3) Linear (L), Quadratic (Q), and Product (P); (4): Quadratic (Q), Hinge (H), and Product (P); (5): Linear (L), Quadratic (Q), and Hinge (H); (6) Linear (L), Quadratic (Q), Hinge (H), and Product (P), and (7) Quadratic (Q), Hinge (H), Product (P), and Threshold (T). The ENMeval package was implemented in R 3.2.4. “Checkerboard2” was employed to search the lowest delta value for Akaike’s information criteria corrected for small sample sizes (AICc) to run the ultimate MaxEnt software among candidate models. The results are shown in Appendix A. A parametrization of RM of 2 and a LQHPT FC were used to execute the MaxEnt model. The “fade-by-clamping” was set in MaxEnt to avoid extrapolations that are outside the environmental range of focus species [33]. The logistic output of MaxEnt was implemented in the final model. A “10th percentile training presence threshold” value (provided by MaxEnt) was used to define the suitable and unsuitable habitats for *D. vitifoliae* [34]. The same threshold was set to define binary presence/absence maps for focus species. A 10-fold cross-validation was employed to execute the final model based on the distribution sites and environmental factors. Future potential distribution map for *D. vitifoliae* was also created by averaging the result of models that were obtained from the three future climate scenarios. Moreover, the potential distribution ranges of suitable map were divided into four levels based on the value of “10th percentile training presence threshold”: unsuitable habitat (0-threshold), low habitat suitability (threshold-0.4), moderate habitat suitability (0.4–0.6), and highly habitat suitability (0.6–1).

### 2.4. Model Evaluation

Evaluation of model performance is an important step in niche modeling [35]. The model generated was evaluated usually through calculating the AUC (area under the curve) of the receiver operating characteristic (ROC) plot [36]. However, some disadvantages were found in the current study due to equal weighting of omission and commission errors, and even AUC value could not provide effective information on the spatial distribution of model errors [37]. The partial ROC metric method (pROC) of Peterson et al. (2008) [38] was employed to assess the performance of the model. The pROC was implemented through the NicheToolbox site with 1000 replicates and E = 0.05 (http://shiny.conabio.gob.mx:3838/nichetoob2/, accessed on 25 November 2020).

### 2.5. Viticultural Regions of World

The suitability of grape phylloxera in most of the regions is explained by the presence of viticulture providing host plants and field surveyed due to the interest of phylloxera monitoring. Thus, the viticultural regions also affected the potential distribution of the *D. vitifoliae*. The viticultural regions used in the current study were from a past study [39,40].

## 3. Results

### 3.1. Species Map and Model Performance for Invasive Species

The list and distribution map of *D. vitifoliae* are illustrated in Figure 1 and Appendix A. Partial ROC evaluation indicated that the potential distribution model of the *D. vitifoliae* is reliable and predictions were statistically significantly better than random (AUC at 0.05 is 0.846 and *p* < 0.001) (Appendix A). A ‘10th percentile training presence logistic threshold’ value of 0.3406 was obtained for this invasive species.

### 3.2. Key Environments Factors 

Among the seven environmental variables in the model, Isothermality (Bio2/Bio7) (*100) (Bio3) and Precipitation of coldest quarter (Bio19) were major factors, influencing the model performance by 33.4 and 28%, respectively (Table 2). Other environmental variables were Annual precipitation (Bio12), Altitude (Alt), and Mean temperature of wettest quarter (Bio8) which contributed to the potential distribution model by 20.5, 11, and 5.6%, respectively. In addition, Mean diurnal range (mean of monthly (max temp–min temp)) (Bio2) and Precipitation of warmest quarter (Bio18) also affected the model with smaller contributions.

### 3.3. Current Invasive Pattern

The present-day potential distribution area of *D. vitifoliae* based on current environmental variables and distribution sites is shown in Figure 2. The map shows that suitable areas cover most areas of Europe, East and North of China, Japan, East of USA, Uruguay, and Southeast South America. Highly suitable areas for this invasive species are also present in the above regions, but their range is smaller than these regions in area. In particular, highly suitable areas are significantly smaller than total suitable areas in Europe. However, the highly suitable area distributions are in the west of America, East of America, and Southeast of Canada in the North America continent. In Europe, the highly suitable areas were scattered in the following countries: Western Ireland, Western France, most areas of Portugal, around of Mediterranean, the Netherlands, Western Germany, Western Turkey and Croatia, etc. In Asia, highly suitable areas focused on eastern China, South Korea, most areas of Japan, and Southeastern India. In addition, Western Australia has some areas with highly suitable areas. The total or potentially suitable habitat area for the pest under current environmental variables was projected to be 1.94 × 108 km^2^. With these areas, approximately 8.56 × 106 km^2^ (~4.41% of total suitable area) exhibited high habitat suitability.

The viticultural regions were mainly present in North America, European, Australia, and Asia according to the result of past studies. Combined with the prediction map of the current model, the potential distribution range of *D. vitifoliae* is as follow: (1) In Europe, the high suitability region focus on Eastern Spain, Eastern France, Eastern England, Northwestern Germany, scattered in Italy, Greece and Hungary, Eastern Turkey, and so on; (2) in North America, most high suitability areas concentrated in the eastern United States and a minor presence was found in western North America; (3) In Asia, most of the high suitability areas are present in the center of China and Japan; (4) In Australia, high suitability areas are located in the Southwest and southeast Australia. Moreover, some high suitability regions are located in the center of Chile and Southern Brazil.

The response curves (Figure 3) created by MaxEnt suggested a high probability of occurrence of the invasive species in regions with the mean diurnal range (Bio2) of 2 to 20 °C, isothermality (Bio3) of 2 to 3.8, mean temperature of wettest quarter (Bio8) of 11 to 34 °C, annual precipitation (Bio12) of 500 to 1800 mm, precipitation of warmest quarter (Bio18) below 500 mm, precipitation of coldest quarter (Bio19) of 200 to 700 mm, and altitude below 700 m.

### 3.4. Future Invasive Pattern

The potential distribution patterns of grape phylloxera under future climate scenarios are illustrated in Figure 4 and Table 3. The total potential distribution area under future climate conditions reduced relative to the suitable area in the current climate condition. However, the highly suitable area significant increased in all future climate conditions.

The area of potential distribution range decreased to ~1.90 × 108 km^2^ (a decrease of 2.1% from the current extent) based on RCP 2.6-2050. The area of highly suitable habitat expanded to ~9.96 × 106 km^2^ (an increase of approximately 16.3% over the current highly suitable area). In addition, the highly suitable areas were increased in some regions under RCP 2.6-2050, for example, Southeast Canada. Similarly, the potential distribution area of total suitable habitats would decrease to approximately ~1.87 × 108 km^2^ (a decrease of 3.6% from the current extent) based on RCP 8.5-2050. In addition, the area of high suitability habitats increased in some regions to approximately ~1.11 × 10^7^ km^2^ (an increase of ~22.8% compared to the current range) (Figure 4 and Table 3). The increased area of high suitability habitats was mainly concentrated in Europe and Asia.

Under the RCP2.6-2070 climatic scenario, MaxEnt predicted a gain in the area of suitable habitat of ~1.90 × 107 km^2^ (a decrease of ~2.1% compared to the current range) (Figure 4 and Table 3). However, the highly suitable area notably increased by 18.4% under this scenario, expanding to about ~1.05 × 107 km^2^. Under RCP 8.5-2070, the model-predicted suitable habitat area was ~1.82 × 107 km^2^, which represents a decrease of approximately 6.5% over the current suitable habitat area (Figure 4 and Table 3). However, the highly suitable area expanded to about ~1.09 × 107 km^2^ (a notable increase of approximately 21.5% over the current highly suitable area) under this scenario. The increased areas also occur in North America, Europe, and Asia.

## 4. Discussion

The current research is the first that has been undertaken to evaluate the potentially suitable area for the invasive pest *D. vitifoliae* under climate change on a global scale by MaxEnt. The MaxEnt model could avoid commission errors in predicting species distribution [41]. This model was used in many previous studies on a global scale, such as: *Cydia pomonella* [42], *Dalbulus maidis* [19], and *Bactrocera zonata* [43]. The final model was considered to be robust and its performance was excellent according to a pROC evaluation. *D. vitifoliae* has been regarded as the most noxious pest of the economically important grape in some parts the world [7]. This invasive pest is native to North America and is considered a potential pest for most grape producing area. The suitable habitat predicted by MaxEnt in the current study is wider than the present distribution of the species. Global economic and quarantine measures for the pest are urgent, due to the expansion of its potential distribution range, especially for Asia, North America, and Europe.

As climate changes, the areas with extreme temperatures or rainfall are expected to either increase or decrease in areas unsuitable for *D. vitifoliae*. Our future potential distribution model showed that climatic changes would significantly affect the distribution model of this invasive species, especially highly suitable area. Our results suggest that there would be an increase in all suitable habitats which include low, moderate, and high areas in response to global warming. This result is in agreement with other studies, such as those on onion maggot (*Delia antiqua*) [44] and *Leucanthemum vulgare* [12]. The model of area increase predicted for *D. vitifoliae* can be explained with changes in increasing temperature under climate change [45]. Therefore, it is necessary to strengthen the quarantine measures for this pest with highly suitable area, such as: China, USA, and most countries of Europe.

The results of the MaxEnt model suggest that temperature and rainfall are the most important variables affecting the current potential distribution map of *D. vitifoliae* among all environmental variables. Isothermality (Bio2/Bio7) (*100) (Bio3) is the most important climatic variable that defines the current potential distribution of this pest. The response curve showed that the probability of the presence of *D. vitifoliae* was very low when the mean temperature of the wettest quarter was outside 11–34 °C. A previous study suggested that the optimum temperature for the growth and development of grape phylloxera is 18–23 °C [46]. When the temperature was below 18 °C, the survival rate of nymphs dropped, and when the temperature was higher than 23 °C, a large number of grape phylloxera died.

The potential distribution of phylloxera in the current study was adequate. The MaxEnt model was implemented on occurrence information and corresponding environmental variables based on the theory of maximum entropy [47]. In other words, the accuracy of the model based on Maxent is mainly affected by two factors: occurrence data and environmental variables. Firstly, current study has not collected complete distribution data of the *D. vitifoliae*; for example, although the pest is established in the pacific northwestern United States, the occurrence data were not evaluated in current study. Interestingly, the recent model shows that this area is a highly suitable area for *D. vitifoliae*, which on the other hand confirms the accuracy of our model. Secondly, climate factors are often seen as the main factor driving species distribution at large scales, and most global-scale studies of niche changes in native and exotic species [45,48]. Climate variables are also considered as the main factors influencing the potential distribution range of *D. vitifoliae* at the global scales based on current model and data. The current model is based on part of abiotic factors only, which limits the accuracy of the potential distribution range of species under climate change. Other factors also influence the results of the model, such as background area (also called “landscape of interest” or “study area”), dispersal ability [49], land use [50], and the distribution range of grape (*Vitis* spp.). Third, the suitability of the host can also affect the potential distribution of the pest. Considering the range of hosts, especially under the conditions of climate change, this can be better able to increase the accuracy of the model. Actually, models for the specialist species had stronger performance than generalist species due to specialist’s species, and easily define the requirement of environmental resources and ecological barriers for predictive models [51].

The study shows that the combination of drought and herbivory stress significantly impact the spread of grape phylloxera by influencing the water status and carbon allocation of grapevines under climate change [50]. Besides abiotic stresses such as drought, plant pathogens and other insect pests also impact the host physiology with a significant impact on *D. vitifoliae* population [52]. The current study focuses on leaf-feeding grape phylloxera populations and did not consider root-feeding populations. The current results might be interesting for future studies investigating the distribution of root-feeding phylloxera instead.

In brief, the current study provides important information on the potential distribution of *D. vitifoliae* under climate change and provides a theoretical reference framework for the management of this invasive species, especially in the formulation of quarantine measures.

## 5. Conclusions

The current research is the first study that has been undertaken to estimate the potential distribution of *D. vitifoliae*, an important pest of grape, under climate change conditions. The model proved to be reliable based on the pROC evaluation. The result suggested that the potentially suitable habitat range is far greater than the current distribution under current climate condition. The suitable and highly suitable area will continue to increase under climate change scenarios. The study also suggested that isothermality and precipitation of coldest quarter were defined as the key elements shaping the distribution of the invasive species. The present research provides a reference to governments or farmers for managing and developing policies against the infestation by *D. vitifoliae*.

## Figures and Tables

**Figure 1 insects-12-00347-f001:**
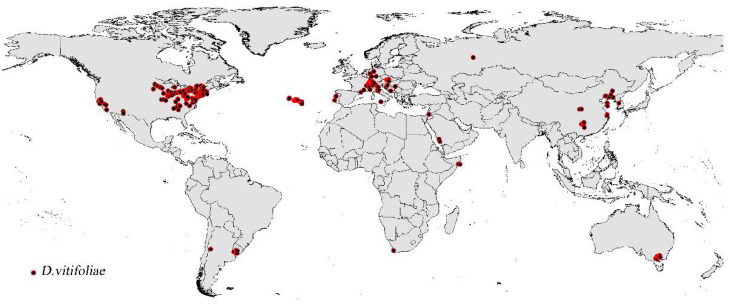
The distribution sites used to simulate the potential distribution of *D. vitifoliae.*

**Figure 2 insects-12-00347-f002:**
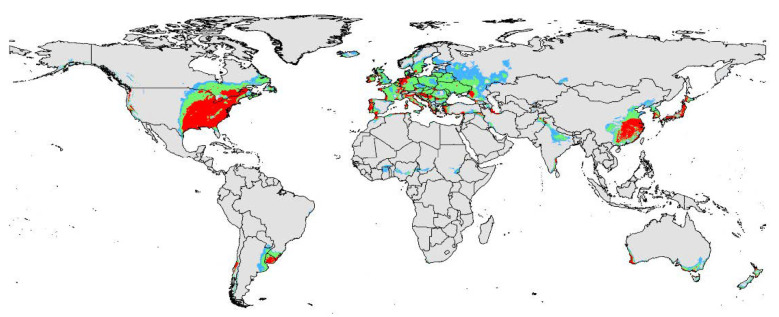
Potential distribution range of *D. vitifoliae* under current climate condition. Gray, unsuitable habitat area; Light blue, low habitat suitability area; Light green, moderate habitat suitability area; Red, highly habitat suitability area.

**Figure 3 insects-12-00347-f003:**
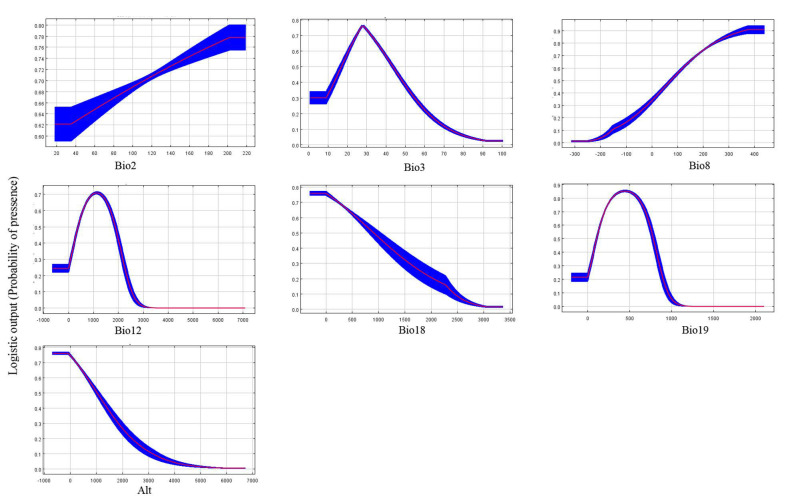
Response curve showing the relationships between the probability of presence of pest and six bioclimatic variables. Values shown are the average over 10 replicate runs; blue margins show ± SD calculated over 10 replicates. For each panel, X axis represents the climate variables or alt, Y axis presents the probability of species occurrence. The classification level of high habitat suitability: 0.6–1. Bio2: mean diurnal range (mean of monthly (max temp-min temp)); Bio3: isothermality (Bio2/Bio7) (*100); Bio8: mean temperature of wettest quarter; Bio12: annual precipitation; Bio18: Precipitation of warmest quarter; Bio19: precipitation of coldest quarter; Alt: altitude.

**Figure 4 insects-12-00347-f004:**
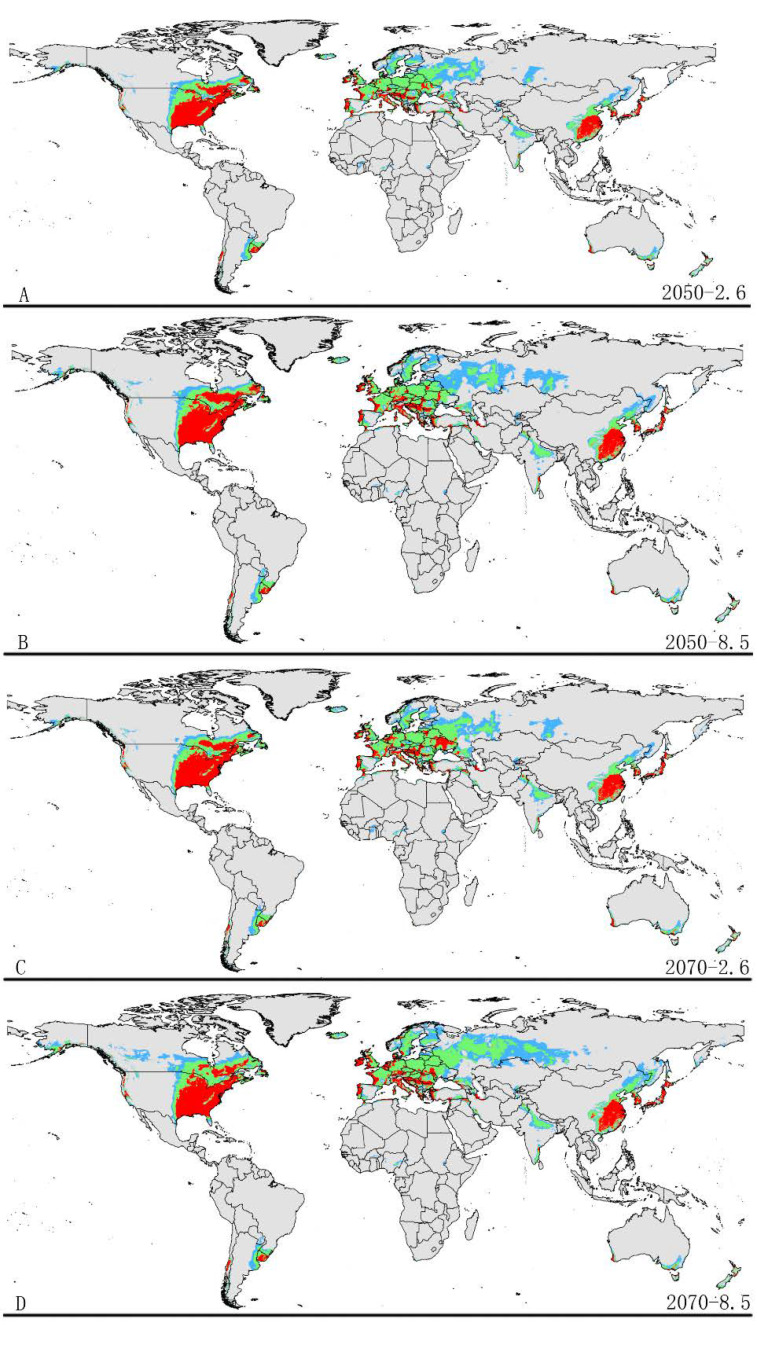
Future species distribution models of grape phylloxera on a global scale under different climate conditions predicated by Maxent. Gray, unsuitable habitat area; Light blue, low habitat suitability area; Light green, moderate habitat suitability area; Red, high habitat suitability area. (**A**) RCP2050-2.6; (**B**) RCP2070-2.6; (**C**) RCP2050-8.5; (**D**) RCP2070-8.5. RCP represents representative concentration pathways; two possible greenhouse gas concentration trajectories (2.6, 8.5). RCP2050-2.6, RCP2070-2.6, RCP2050-8.5, and RCP2070-8.5 represent four climate change scenarios/year.

**Table 1 insects-12-00347-t001:** Climatic variables used as predictor variables to model the potential distribution of *D. vitifoliae.*

Environmental Variables	
Bio2	Mean Diurnal Range
Bio3	Isothermality (BIO2/BIO7) (*100)
Bio8	Mean Temperature of Wettest Quarter
Bio12	Annual Precipitation
Bio18	Precipitation of Warmest Quarter
Bio19	Precipitation of Coldest Quarter
Alt	Altitude

**Table 2 insects-12-00347-t002:** Relative contribution of each climatic variable to MaxEnt model.

Environmental Variables	Relative Contribution
Isothermality (BIO2/BIO7) (*100) (Bio3)	33.4%
Precipitation of Coldest Quarter (Bio19)	28%
Annual Precipitation (Bio12)	20.6%
Altitude (alt)	11%
Mean Temperature of Wettest Quarter (Bio8)	5.6%
Mean Diurnal Range (Bio2)	0.9%
Precipitation of Warmest Quarter (Bio18)	0.6%

**Table 3 insects-12-00347-t003:** Predicted suitable areas worldwide for *D. vitifoliae* under current and future climatic conditions (km^2^).

Classification Level ^a^	Current Climate	Future Climate Conditions			
		RCP2.6-2050	RCP8.5-2050	RCP2.6-2070	RCP8.5-2070
Suitable habitat (0.3406–1)	~1.94 × 10^8^	~1.90 × 10^8^	~1.87 × 10^8^	~1.90 × 10^8^	~1.82 × 10^8^
		(−2.1%) ^b^	(−3.6%)	(−2.1%)	(−6.5%)
Low habitat suitability (0.3406–0.4)	~9.20 × 10^6^	~1.01 × 10^7^	~1.21 × 10^7^	~9.67 × 10^6^	~1.37 × 10^7^
		(8.9%)	(23.9%)	(4.9%)	(32.8%)
Moderate habitat suitability (0.4–0.6)	~1.31 × 10^7^	~1.52 × 10^7^	~1.45 × 10^7^	~1.50 × 10^7^	~1.87 × 10^7^
		(13.8%)	(9.6%)	(12.7%)	(29.9%)
High habitat suitability (0.6–1)	~8.56 × 10^6^	~9.96 × 10^6^	~1.11 × 10^7^	~1.05 × 10^7^	~1.09 × 10^7^
		(16.3%)	(22.8%)	(18.4%)	(21.5%)

^a^ suitable and unsuitable habitat of the grape phylloxera of all suitable distribution areas; ^b^ the brackets show the proportion of the area of the suitable area under future climate scenarios relative to the area of the suitable area under the current climate scenario.

## Data Availability

The data presented in this study are available in Appendix A.

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
