# Peer review of "Potential Global Distribution of Daktulosphaira vitifoliae under Climate Change Based on MaxEnt"

_insects, 2021, doi:10.3390/insects12040347_

Round 1
Reviewer 1 Report
The manuscript Insects-1083998 on distribution of grape phylloxera is a pretty good manuscript. It does require some English editing as some of the wording is awkward, but generally, the paper is quite understandable and well written. I just have a couple of important comments, and then a number of minor editorial type comments.
- On line 85 you say 128 sites, but on line 92 you say 784 locations remained. Is the first number an error or typo? I cannot tell as I do not have access tot eh supplemental files.
- Grape phylloxera is established in the pacific northwestern United States, but I do not see this area included in Figure 1. Was this included in your analysis? Interesting to note that the model does predict high probability of occurrence in this area.
- Line 106 – you use correlation to eliminate collinear variables. With paired correlations, however, how do you select the one to eliminate? There must be more of a decision here. In other words, If A and B are correlated, how do you decide to eliminate B and not A? If you did this based on ecological knowledge fine, but you should state the criteria fully.
Minor changes:
Line 12 and elsewhere – change Fitch to (Fitch) – should be in parentheses.
Line 40 – change invasived to moved
Line 52 – management should be manage
Line 56 – “provide some reference” should be reworded as this is unclear what you mean.
Line 58 – “change” is used twice in the sentence – reword
Line 61 – change “make the” to “develop”
Line 71 and ff – you are not creating the distribution, you are modelling it and identifying climate factors associated with it. In this paragraph, you should also indicate that you use the model to examine distribution under future climate scenarios.
Line 98 – change with to and
Line 104 change lose to reduce.
Line 130 “that can be used”
Line 204 – italicize the latin name
Line 343 – provide latin names and taxonomic authorities for both species mentioned.
Author Response
Reply to Reviewer 1
Comment 1:On line 85 you say 128 sites, but on line 92 you say 784 locations remained. Is the first number an error or typo? I cannot tell as I do not have access tot eh supplemental files.
Response to comment 1:Thanks for your value and careful work. This is our written error. We changed 784 to 103 in revised manuscript.
Comment 2: Grape phylloxera is established in the pacific northwestern United States, but I do not see this area included in Figure 1. Was this included in your analysis? Interesting to note that the model does predict high probability of occurrence in this area.
Response to comment 2:Thanks for your value work and value suggestion. Sorry, it is our mistake. Actually, studies shown that the grape phylloxera native to eastern united states. Our data not include the data that from northwestern United states. Although there are some distributions in the southwest, the results of the model show that this area may not be a highly suitable area. Because the model is affected by many factors, we only consider climate and terrain factors. Future evaluations should add other conditions, such as light and other conditions.
Comment 3: Line 106 – you use correlation to eliminate collinear variables. With paired correlations, however, how do you select the one to eliminate? There must be more of a decision here. In other words, If A and B are correlated, how do you decide to eliminate B and not A? If you did this based on ecological knowledge fine, but you should state the criteria fully.
Response to comment 3:Thanks for your value suggestion. In current studies, Pearson’s correlation was implemented in SPSS statistics. This approach was used in many previous studies, such as: Xu et al., 2019; Low et al., 2020; Tang et al., 2018 and so on.
Xu DP, Zhuo ZH, Wang RL, Ye M, Pu B. 2019. Modeling the distribution of Zanthoxylum armatum in China with MaxEnt Modeling. Global ecology and conservation, 19: e00691.
Low BW, Zeng YW, Tan HH, Yeo DCJ. 2020. Predictor complexity and feature selection affect Maxent model transferability: evidence from global freshwater invasive species. Diversity and distributions (online).
Tang JH, Li JH, Lu H, Lu FP, Lu BQ. 2018. Potential distribution of an invasive pest, Euplatypus parallelus, in China as predicted by Maxent. Pest management science, 75(6): 1630-1637.
Comment 4: Line 12 and elsewhere – change Fitch to (Fitch) – should be in parentheses.
Response to comment 4:Thanks for your value work. We changed it according reviewer’s suggestion.
Comment 5: Line 40 – change invasived to moved
Response to comment 5:Thanks for your careful work. We changed it in revised manuscript.
Comment 6: Line 52 – management should be manage
Response to comment 6:Thanks for your careful work. We changed it in revised manuscript.
Comment 7: Line 56 – “provide some reference” should be reworded as this is unclear what you mean.
Response to comment 7:Thanks for you value suggestion. We change this sentence in revised manuscript.
Comment 8: Line 58 – “change” is used twice in the sentence – reword
Response to comment 8:Thanks for your value suggestion, we change it with “influence” in revised manuscript.
Comment 9: Line 61 – change “make the” to “develop”
Response to comment 9:Thanks for your careful work. We changed it in revised manuscript.
Comment 10: Line 71 and ff – you are not creating the distribution, you are modelling it and identifying climate factors associated with it. In this paragraph, you should also indicate that you use the model to examine distribution under future climate scenarios.
Response to comment 10:Thanks for your careful work and value suggestions. We changed it in revised manuscript.
Comment 11: Line 98 – change with to and
Response to comment 11:Thanks for your careful work and value suggestions. We changed “with” to “used” in revised manuscript.
Comment 12: Line 104 change lose to reduce.
Response to comment 12:Thanks for your value suggestions. We changed “lose” to “reduce” in revised manuscript.
Comment 13: Line 130 “that can be used”
Response to comment 13:Thanks for your value suggestions. We changed it in revised manuscript.
Comment 14: Line 204 – italicize the latin name
Response to comment 14:Thanks for your careful work. We changed it in revised manuscript.
Comment 15: Line 343 – provide latin names and taxonomic authorities for both species mentioned.
Response to comment 15:Thanks for your careful work. We provide latin names in revised manuscript.
Reviewer 2 Report
The authors Ji et al. describe an interesting study about the global distribution of leaf-feeding grape phylloxera populations and their influencing climatic and topographic factors. The study presents important results for scientists and growers dealing with plant feeding parasites. Knowledge about the climatic factors triggering phylloxera spread and survivorship could contribute to the development of novel pest management approaches.
The general writing style is acceptable, but could be improved by cross reading of a native English speaker. Please be consistent with taxonomical names (spellings, italics) and the tempus through the manuscript.
Although I am not an expert in pest modelling approaches, the presented analytical approach is sounding to me. However, I do feel that the distribution of the Vitis host poses a significant bias to the global distribution of grape phylloxera, here explained by abiotic factors. In particular Figure 1 shows that the main driver for global grape phylloxera distribution is the presence of viticulture, providing suitable Vitis hosts and observations/data due to its pest status in these regions. The red points of the map match with common viticultural regions in South America, South Africa, Europe (incl. Azores), East Asia and Australia. In the US the red spots mark the viticulture area in California and New York combined with the natural distribution of the main Vitis hosts (V. riparia, V. rupestris, V. vulpina). Please consider that phylloxera larvae in first place need a suitable Vitis host providing the nutritional basis for development and reproduction, before they are affected by climatic factors. As an idea and to overcome this bias you could analyse subregions (e.g. Europe, China, Western US), defined by an equal host species distribution, and correlate the insect population data with the climate/topographic data within this regions on a smaller scale.
Because of the novelty and importance of the presented study, I do support the publication of the submitted manuscript with major revisions.
Minor comments:
Abstract
- Line 19: replace “developed” with “investigated”
Introduction
The introduction history of grape phylloxera is redundant in the manuscript.
- Line 30: Please be precise: Phylloxera`s leaf feeding habit predominantly depends on the Vitis host genotype rather than on the geographical location. Phylloxera does commonly feed on leaves of rootstocks in Eurasia as well as on vinifera roots in the US.
- Line 35: Replace “allow” by “facilitate”
- Line 38: Human cultivation actions are probably the major cause of phylloxera spread between vineyards. However, mobile L1 stages or winged Alates should be mentioned as further possibilities.
- Line 39: Grape phylloxera ( vitifoliae) and the grape mealybug (P. maritimus) are different insect species. “Mealybug” is one word.
- Line 43-50: Irrelevant information
- Line 54: Nowadays most commercial rootstocks are rated tolerant against grape phylloxera. This means, they host minor but reproducing phylloxera populations on the roots without causing visible damage symptoms or severe yield losses to the host. Therefore “resistant rootstocks” is no suitable. Please provide more detailed information here. See: Forneck et al. 2016 and Powell and Korosi 2013.
- Line 73: “In addition”
Material and Methods
- Line 89: “kept”
- Line 97: Name or provide a table with the used variables
- Lin 98: Do you mean “and considered” instead of “with considered”?
- Line 131: “range” instead or “rage”
- Lines 147 – 156: Some word appear in a different writing style: “used” line 147; “employed” “distribution sites” line 153/154, also on the further manuscript.
Results:
- Line 204: “Daktulosphaira vitifoliae” to be spelled in italics
Discussion
- Table 1: Should be moved to the Results section and include a sort of significance measure e.g. the correlation coefficient
- Line 343: Use consistent taxonomical nomenclatures for insect names through the manuscript (trivial or taxonomic, if taxonomic use italics)
- Table 2: Should be moved to the Results section. The table is not easy to understand. Maybe you could modify it or explain details in the table legend.
- It should be discussed that the presented study focused on the climate conditions influencing leaf feeding grape phylloxera populations. However, phylloxera root-feeding populations, which are generally more harmful to the host, were not considered (e.g. by soil humidity or temperature parameters) A section about similar studies prediction the distribution areas of other insect species concerning trends and applications would be a benefit for the study.
- A discussion about abiotic stresses (e.g. droughts) and their implications for host defences, might be interesting (Savi et al. 2019 or Powell et al 2013)
- A section about similar studies prediction the distribution areas of other insect species concerning trends and applications would be a benefit for the study.
References
- Line 427: “Daktulosphaira” not “Daktulospharia”
- Line: 429 “Eitle” not “Eitie”; “Daktulosphaira” not “Daktulospharia”
- Line 432: “Daktulosphaira” not “Daktulospharia”
References stated by the reviewer
- Forneck, A., Powell, K. S., & Walker, M. A. (2016). Scientific opinion: improving the definition of grape phylloxera biotypes and standardizing biotype screening protocols. American Journal of Enology and Viticulture. 67(4):371-376
- Powell, K. S., & Korosi, G. A. (2013). 'Taking the strain'-selecting the right rootstock to protect against endemic phylloxera strains. Acta Horticulturae 1045:99-107
- Powell, K. S., Cooper, P. D., & Forneck, A. (2013). The biology, physiology and host–plant interactions of grape phylloxera Daktulosphaira vitifoliae. Advances in insect physiology, 45, 159-218.
- Savi, T., González, A. G., Herrera, J. C., & Forneck, A. (2019). Gas exchange, biomass and non-structural carbohydrates dynamics in vines under combined drought and biotic stress. BMC plant biology, 19(1): 1-11.
Author Response
Reply to Reviewer 2
Comment 1: Although I am not an expert in pest modelling approaches, the presented analytical approach is sounding to me. However, I do feel that the distribution of the Vitis host poses a significant bias to the global distribution of grape phylloxera, here explained by abiotic factors. In particular Figure 1 shows that the main driver for global grape phylloxera distribution is the presence of viticulture, providing suitable Vitis hosts and observations/data due to its pest status in these regions. The red points of the map match with common viticultural regions in South America, South Africa, Europe (incl. Azores), East Asia and Australia. In the US the red spots mark the viticulture area in California and New York combined with the natural distribution of the main Vitis hosts (V. riparia, V. rupestris, V. vulpina). Please consider that phylloxera larvae in first place need a suitable Vitis host providing the nutritional basis for development and reproduction, before they are affected by climatic factors. As an idea and to overcome this bias you could analyse subregions (e.g. Europe, China, Western US), defined by an equal host species distribution, and correlate the insect population data with the climate/topographic data within this regions on a smaller scale.
Response to comment 1: Thanks for your value suggestion. Your suggestion is very good. Actually, we will use another model (CLMEX) to study grape phylloxera in next work. The model will include soil parameters, vegetation types and corresponding physiological parameters of this pest. The physiological parameters will include the starting temperature of pupal and larvae development, the effective accumulated temperature of this pest and so on. Your suggestions gave us good ideas, such as considering nutritional preferences and analyzing the distribution of subregions separately, especially for consider root-feeding form. We will fully consider your opinion in the next step. MaxEnt has its own shortcoming, for example lack the physiological parameters in software setting. Thus, in current study, we just considered the climatic and topological variable.
Comment 2: Line 19: replace “developed” with “investigated”
Response to comment 2: Thanks for your value suggestion. We change it in revised manuscript.
Comment 3: Line 30: Please be precise: Phylloxera`s leaf feeding habit predominantly depends on the Vitis host genotype rather than on the geographical location. Phylloxera does commonly feed on leaves of rootstocks in Eurasia as well as on vinifera roots in the US.
Response to comment 3: Thanks for your value suggestion. According your suggestion, we rewrite this sentence in revised manuscript.
Comment 4: Line 35: Replace “allow” by “facilitate”
Response to comment 4: Thanks for your value suggestion. We change it in revised manuscript.
Comment 5: Line 38: Human cultivation actions are probably the major cause of phylloxera spread between vineyards. However, mobile L1 stages or winged Alates should be mentioned as further possibilities.
Response to comment 5: Thanks for your value suggestion. We change it in revised manuscript.
Comment 6: Line 39: Grape phylloxera ( vitifoliae) and the grape mealybug (P. maritimus) are different insect species. “Mealybug” is one word.
Response to comment 6: Thanks for your careful work. Sorry for our mistake, we correct it in revised manuscript.
Comment 7: Line 43-50: Irrelevant information
Response to comment 7: Thanks for your value suggestion. We deleted this information in revised manuscript.
Comment 8: Line 54: Nowadays most commercial rootstocks are rated tolerant against grape phylloxera. This means, they host minor but reproducing phylloxera populations on the roots without causing visible damage symptoms or severe yield losses to the host. Therefore “resistant rootstocks” is no suitable. Please provide more detailed information here. See: Forneck et al. 2016 and Powell and Korosi 2013.
Response to comment 8: Thanks for your value suggestion. We added this information in revised manuscript.
Comment 9: Line 73: “In addition”
Response to comment 9: Thanks for your careful work. We change it in revised manuscript.
Comment 10: Line 89: “kept”
Response to comment 10: Thanks for your careful work, we change it in revised manuscript.
Comment 11: Line 97: Name or provide a table with the used variables
Response to comment 11: Thanks for your careful work, we provide a table in revised manuscript.
Comment 12: Lin 98: Do you mean “and considered” instead of “with considered”?
Response to comment 12: Thanks for your careful work, we change it in revised manuscript.
Comment 13: Line 131: “range” instead or “rage”
Response to comment 13: Thanks for your careful work, we change it in revised manuscript.
Comment 14: Lines 147 – 156: Some word appear in a different writing style: “used” line 147; “employed” “distribution sites” line 153/154, also on the further manuscript.
Response to comment 14: Thanks for your careful work, we change it in revised manuscript.
Comment 15: Line 204: “Daktulosphaira vitifoliae” to be spelled in italics
Response to comment 15: Thanks for your careful work, we change it in revised manuscript.
Comment 16: Table 1: Should be moved to the Results section and include a sort of significance measure e.g. the correlation coefficient
Response to comment 16: Thanks for your careful work and value suggestion. We moved the Table 1 to Results section. In additional, the correlation coefficient was present in Table S3.
Comment 17: Line 343: Use consistent taxonomical nomenclatures for insect names through the manuscript (trivial or taxonomic, if taxonomic use italics)
Response to comment 17: Thanks for your careful work, we change it in revised manuscript.
Comment 18: Table 2: Should be moved to the Results section. The table is not easy to understand. Maybe you could modify it or explain details in the table legend.
Response to comment 18: Thanks for your careful work and value suggestion. We moved the Table 1 to Results section and add explain in notes.
Comment 19: It should be discussed that the presented study focused on the climate conditions influencing leaf feeding grape phylloxera populations. However, phylloxera root-feeding populations, which are generally more harmful to the host, were not considered (e.g. by soil humidity or temperature parameters). A section about similar studies prediction the distribution areas of other insect species concerning trends and applications would be a benefit for the study.
Response to comment 19: Thanks for your value suggestion. We added this section in discussion.
Comment 20: A discussion about abiotic stresses (e.g. droughts) and their implications for host defences, might be interesting (Savi et al. 2019 or Powell et al 2013)
Response to comment 20: Thanks for your value suggestion. We added this section in discussion.
Comment 21: A section about similar studies prediction the distribution areas of other insect species concerning trends and applications would be a benefit for the study.
Response to comment 21: Thanks for your value suggestion. We added this section in discussion.
Comment 22: Line 427: “Daktulosphaira” not “Daktulospharia”
Response to comment 22: Thanks for your careful work, we change it in revised manuscript.
Comment 23: Line: 429 “Eitle” not “Eitie”; “Daktulosphaira” not “Daktulospharia”
Response to comment 23: Thanks for your careful work, we change it in revised manuscript.
Comment 24: Line 432: “Daktulosphaira” not “Daktulospharia”
Response to comment 24: Thanks for your careful work, we change it in revised manuscript.
Round 2
Reviewer 2 Report
The authors Ji et al. describe an interesting study about the global distribution of leaf-feeding grape phylloxera populations and their influencing climatic and topographic factors. The study presents important results for scientists and growers dealing with plant feeding parasites. Knowledge about the climatic factors triggering phylloxera spread and survivorship could contribute to the development of novel pest management approaches.
The manuscript and the writing style improved significantly after the first round of review. However, there are some minor issues that need to be addressed. The manuscript would profit by proof reading from a native English speaker. The bias regarding the host plant distribution for the correlation of phylloxera leaf population occurrences to climate factors on global scale was not addressed yet, which is my major concern regarding the data basis. Because it seems that the previously suggested analysis of subregions (with equal host distribution) cannot be adopted, it is necessary to be careful and critical in particular when talking about phylloxera distribution on global scale. I added some comments on the following.
Because of the novelty and importance of the presented study, I do strongly support the publication of the submitted manuscript after another round of revision(s). In the following I added major and minor comments and suggestions along the manuscript:
Simple Summary and Abstract
Major points:
(1) Australia was not named on the list of countries, although there are several publications reporting phylloxera leaf populations there.
(2) It should be pointed out clearly in the beginning that the analysis was done regarding leaf populations.
Minor points:
Line 12: Rephrase: “widely occurrence in” with “present in” or “widely spread across”
Line 13: Add: In the current study
Lines 13/14: Check: environmental
Line 14: Rephrase: “created“ (e.g. “analysed “or “predicted” or “investigated”)
Line 14: Rephrase: “highly suitable ranges”
Line 15: Change: “East and North of China” to “East and North China”
Line 15: Change: “East of USA” to “the Eastern USA”
Line 15: Change: “Southeast South America” to “the Southeast of South America”
Lines 15/16: Last sentence is unclear
Line 17: Omit: “The”
Line 18: Omit: “both”
Line 19: Add :“based on population occurrence data”
Line 20: Omit: comma after conditions
Lines 20/21: Adapt region names as mentioned above
Line 21: Change “are the primary potentially suitable areas” to “are suitable areas for the occurrence of phylloxera leaf populations”
Line 21: Omit: “also”
Line: 22: Rephrase: what do you mean by coldest quarter? Maybe “during the winter/dormancy period”?
Line 22: Check: “explained/accounted for more than 60% of the predicted/modeled phylloxera occurrence on leaves”
Line 23: Change: “will” to “provide important knowledge for”
Line 23: Change: “… governmental decision makers to develop pest control and management strategies against D. vitifoliae.
Keywords:
Minor points:
Line 25: “Global” and “potential distribution” might not be good key words
Introduction:
Major points
Lines 30-32: Incorrect: “which feeds on feed on leaves of rootstocks in Eurasia as well as on vinifera roots in the US [3]. Interestingly, grape phylloxiera feeds mostly on the leaves in 31 its native range and predominantly on the roots in Eurasia [4]. This”; the host preference of the phylloxera biotype is not bound to the geographical location.
Lines 51-53: The concept here is still unclear. Nowadays most commercial rootstocks are rated as tolerant against grape phylloxera. This means, they do host minor but reproducing phylloxera populations without causing visible damage symptoms or severe yield losses to the host. Therefore “resistant rootstocks” is not suitable. Please present this information correctly and in a clear way.
Lines 62-64: Critical statement: There are phylloxera field studies that analyzed insect distribution e.g based on across geographical locations, differing soils (e.g. Pfeffer 2005), climate factors (e.g. Arancibia et al. 2018) and management techniques (e.g. Arancibia et al. 2019, Lotter et al. 1999)
Minor points
Line 30: Change: “which feeds” to “feeding”
Lines 32-35: Repetitive information
Lines 35-37: Rephrase: “Both forms of root damage facilitate soil-borne pathogens to enter into the root cortex leading to/resulting in growth reductions, yield loss or ultimate vine death [3,5].”
Line 39: Modify: “planting material“ to „plant material“
Lines 39/40: Incomplete sentence
Line 40: Check and be precise: “probably the Eastern part” there is scientific evidence for the native range of grape phylloxera.
Line 41: Add: “Most viticultural countries”
Lines 40-44: Repetitive information
Line 46: Check: “Vine mealybug” wrong species
Line 48: Add: “resistance and tolerance mechanisms”
Line 50: Two sentences: … “management of grape phylloxera. However…”
Line 56: Rephrase: “Therefore _ alternative phylloxera surveillance methods, models to predict phylloxera occurrence and strict quarantine measures might prevent the spread of the pest into regions or countries where grape phylloxera has not yet been recorded.”
Line 56: Do you mean the “climate change” or in general “climatic changes”
Line 56: Rephrase: through changes” to e.g. “by affecting”
Line 57: Check: “growth rates”
Line 57: Specify: “reproductive pressure” and “transmission”
Lines 57/58: Rephrase: “significantly affecting their _distribution.”
Line 60: Rephrase: “expansion of grape phylloxera”
Line 62: Omit: “some”
Line 63: Omit: “have”
Line 70: Check: “global” twice the word
Line: 73: Change: “policies for the management and control of pest.” to “policies for pest management and control.”
Material and Methods
Major points
Lines 131-135: Irrelevant information for the M&M sections. Could be moved to discussion or taken out.
Minor points
Line 162: “value” different writing style
Results
Major points
Lines 187-202: The suitability of grape phylloxera in most of the named regions is primarily explained by the presence of viticulture providing host plants and observations/reports due to the interest of phylloxera monitoring. If previously published phylloxera field observations (Supplement) serve as data basis for the presented study, they bias the conducted analysis. Therefore, for me, it is not correct to link the geographical distribution areas of D. vitifoliae in current viticultural areas with the analyzed climate data only, as done here (current invasive patterns). However, I would agree to apply your results/models to pyhlloxera distribution in potentially novel distribution areas without a current viticulture (lines 196-198, future invasive patterns). I suggest to be very clear in this point and maybe combine and rewrite parts of the two paragraphs. As mentioned in the previous round of comments, for linking climate factors and phylloxera distributions within traditional viticultural regions, a homogeneous host distribution in the analyzed areas/subregions is needed.
Lines 190-193: Please check this part. It is not clear to me. What do you mean by highly suitable? How is this defined?
Lines 200-201: What do you mean with “ … suitable habitat area […] was projected to be 1.94×108 km2.”? In particular it is not clear how you correlate area units (square km) to habitat suitability. Therefore for me Table 3 (including the legend) is not self explaining.
Minor points
Figure 1: Figure 1 is first mentioned in the Material and Methods section. Therefore it should be moved there or be mentioned in the Results section. Check that all other figures are placed after the paragraph they are named firstly through the manuscript.
Line 200: km2 should be correct to km2 or km-2 according to the journal unit requirements.
Table 3: The first column (pest) might not be necessary.
Figure 3: Is not self explaining. Axes units and the parameters are missing are missing.
Lines 230/232: Twice “However,”
Discussion
Major Points
Line 336-337: Climate factors are undoubtly important for the distribution of insects (pest) species. However, in case of specialist species (such as D. vitifoliae) the presence of a matching Vitis spp. host is essential in first place. I suggest to differentiate between generalist and specialist or milder the statement here.
Line 434: Add: “The current study focuses on leaf-feeding grape phylloxera populations and did not consider root-feeding populations.”
Lines 343-345: Omit, because it is not done yet:” However, study suggest that root-feeding form cause most economic damage [12]. Thus, these 343 factors, including soil humidity and temperature, grape phylloxera form, especially for host-plant 344 availability [14] were considered in future study.” You might add a sentence that your results/factors might be interesting/groundbreaking for future studies investigating the distribution of root-feeding phylloxera instead.
Minor Points
Line 326: Omit: “These studies all shown good simulation results.”
Line 330-332: Check and rephrase more precisely: “However, many biotic factors, such as interspecific interactions [49] (e.g. inter- or intra species competition and 331 propagule pressure) also affect the potential range of species [49].”
Line 333: Omit: “also”
Line 333: Add: “grapevine hosts (Vitis spp.)
Line 334: Omit one comma
Line 335: Unformatted citation
Line 341: The use of leaf galling phylloxera only must should be mentioned from the beginning (e.g. Abstract and Material and Methods), to be clear.
Line 434: Add: “The current study focuses on leaf-feeding grape phylloxera populations and did not consider root-feeding populations.”
Line 340: Modify: “In additional, beside the drought stress and insect 340 pests, plant pathogens, also impact the physiology and health on vine [53].” to “Besides abiotic stresses such as drought, plant pathogens and/or other insect pests impact the host physiology with significant impact on D. vitifoliae populations [53].”
Author Response
Response to reviewer: Comment 1: The bias regarding the host plant distribution for the correlation of phylloxera leaf population occurrences to climate factors on global scale was not addressed yet, which is my major concern regarding the data basis. Because it seems that the previously suggested analysis of subregions (with equal host distribution) cannot be adopted, it is necessary to be careful and critical in particular when talking about phylloxera distribution on global scale. I added some comments on the following. Response to comment 1: Comment 2: Australia was not named on the list of countries, although there are several publications reporting phylloxera leaf populations there. Response to comment 2: Thanks for your careful work. Indeed, the Australia is a distribution area for phylloxera, however, our result suggested that Australia is not a highly suitable region. It is a moderate habitat suitable region. The suitable region divided into four level: highly suitable, moderate suitable, low suitable and unsuitable region. We just list some region or countries that the pest could be occur with highly possibility. So, Australia was not list in Simple Summary and Abstract. Comment 3: It should be pointed out clearly in the beginning that the analysis was done regarding leaf populations. Response to comment 3: Thanks for your value suggestion, we add this information in revised manuscript. Comment 4: Line 12: Rephrase: “widely occurrence in” with “present in” or “widely spread across” Response to comment 4: Thanks for your careful work. We correct this according to your suggestion. Comment 5: Line 13: Add: In the current study Response to comment 5: Thanks for your careful work. We correct this according to your suggestion. Comment 6: Lines 13/14: Check: environmental Response to comment 6: Thanks for your careful work. We correct this according to your suggestion. Comment 7: Rephrase: “created“ (e.g. “analysed “or “predicted” or “investigated”) Response to comment 7: Thanks for your careful work. We correct this according to your suggestion. Comment 8: Line 14: Rephrase: “highly suitable ranges” Response to comment 8: The suitable range of this pest divided into four level: highly suitable range, moderate suitable range, low suitable range and unsuitable range. We added these statement in revised manuscript. Comment 9: Line 15: Change: “East and North of China” to “East and North China” Response to comment 9: Thanks for your careful work. We correct this according to your suggestion. Comment 10: Line 15: Change: “East of USA” to “the Eastern USA” Response to comment 10: Thanks for your careful work. We correct this according to your suggestion. Comment 11: Line 15: Change: “Southeast South America” to “the Southeast of South America” Response to comment 11: Thanks for your careful work. We correct this according to your suggestion. Comment 12: Lines 15/16: Last sentence is unclear Response to comment 12: Thanks for your careful work. We correct this according to your suggestion. Comment 13: Line 17: Omit: “The” Response to comment 13: Thanks for your value suggestion. We delete“the”in revised manuscript. Comment 14: Line 18: Omit: “both” Response to comment 14: Thanks for your value suggestion. We delete“both”in revised manuscript. Comment 15: Line 19: Add :“based on population occurrence data” Response to comment 15: Thanks for your value suggestion. We add“population”in revised manuscript. Comment 16: Line 20: Omit: comma after conditions Response to comment 16: Thanks for your value suggestion. We delete“comma”in revised manuscript. Comment 17: Lines 20/21: Adapt region names as mentioned above Response to comment 17: Thanks for your value suggestion. We change it in revised manuscript. Comment 18: Line 21: Change “are the primary potentially suitable areas” to “are suitable areas for the occurrence of phylloxera leaf populations” Response to comment 18: Thanks for your value suggestion. We change it in revised manuscript. Comment 19: Line 21: Omit: “also” Response to comment 19: Thanks for your value suggestion. We delete“also”in revised manuscript. Comment 20: Line: 22: Rephrase: what do you mean by coldest quarter? Maybe “during the winter/dormancy period”? Response to comment 20: Thanks for your careful work, the precipitation of coldest quarter is a name of Bioclim that were defined by IPCC. Comment 21: Line 22: Check: “explained/accounted for more than 60% of the predicted/modeled phylloxera occurrence on leaves” Response to comment 21: Thanks for your careful work. In fact, the model obtained by niche simulation is a probability of occurrence, that is, the probability that a species will occur in a certain area. The most important climate variables may have the greatest impact on the final results of the model, that is, the greatest impact on the possibility of species distribution in a certain area. Comment 22: Line 23: Change: “will” to “provide important knowledge for” Response to comment 22: Thanks for your value suggestion. We change it in revised manuscript. Comment 23: Line 23: Change: “… governmental decision makers to develop pest control and management strategies against D. vitifoliae. Response to comment 23: Thanks for your value suggestion. We change it in revised manuscript. Comment 24: Line 25: “Global” and “potential distribution” might not be good key words Response to comment 24: Thanks for your value suggestion. We change it in revised manuscript. Comment 25: Lines 30-32: Incorrect: “which feeds on feed on leaves of rootstocks in Eurasia as well as on vinifera roots in the US [3]. Interestingly, grape phylloxiera feeds mostly on the leaves in 31 its native range and predominantly on the roots in Eurasia [4]. This”; the host preference of the phylloxera biotype is not bound to the geographical location. Response to comment 25: Thanks for your value suggestion and careful work. We modify this sentence in revised manuscript. Comment 27: Lines 51-53: The concept here is still unclear. Nowadays most commercial rootstocks are rated as tolerant against grape phylloxera. This means, they do host minor but reproducing phylloxera populations without causing visible damage symptoms or severe yield losses to the host. Therefore “resistant rootstocks” is not suitable. Please present this information correctly and in a clear way. Response to comment 27: Thanks for your value suggestion. We rewrite this sentence in revised manuscript. Comment 28: Lines 62-64: Critical statement: There are phylloxera field studies that analyzed insect distribution e.g based on across geographical locations, differing soils (e.g. Pfeffer 2005), climate factors (e.g. Arancibia et al. 2018) and management techniques (e.g. Arancibia et al. 2019, Lotter et al. 1999) Response to comment 28: Thanks for you value suggestion. We added these statement in revised manuscript. Comment 29: Line 30: Change: “which feeds” to “feeding” Response to comment 29: Thanks for your value suggestion. We change it in revised manuscript. Comment 30: Lines 32-35: Repetitive information Response to comment 30: Thanks for your value suggestion. We delete these sentence in revised manuscript. Comment 31: Lines 35-37: Rephrase: “Both forms of root damage facilitate soil-borne pathogens to enter into the root cortex leading to/resulting in growth reductions, yield loss or ultimate vine death [3,5].” Response to comment 31: Thanks for your value suggestion. We rephrase this sentence according to reviewer’s suggestion. Comment 32: Line 39: Modify: “planting material“ to „plant material“ Response to comment 32: Thanks for your value suggestion. We change it in revised manuscript. Comment 33: Lines 39/40: Incomplete sentence Response to comment 33: Thanks for your careful work. We change it in revised manuscript. Comment 34: Line 40: Check and be precise: “probably the Eastern part” there is scientific evidence for the native range of grape phylloxera. Response to comment 34: Yes,you are correct, the native range of this pest is North America. We correct this in revised manuscript. Comment 35: Line 41: Add: “Most viticultural countries” Response to comment 35: Thanks for your value suggestion. We add it in revised manuscript. Comment 36: Lines 40-44: Repetitive information Response to comment 36: Thanks for your value suggestion. We delete it in revised manuscript. Comment 37: Line 46: Check: “Vine mealybug” wrong species Response to comment 37: Thanks for your careful work, we correct it in revised manuscript. Comment 38: Line 48: Add: “resistance and tolerance mechanisms” Response to comment 38: Thanks for your value suggestion. We add it in revised manuscript. Comment 39: Line 50: Two sentences: … “management of grape phylloxera. However…” Response to comment 39: Thanks for your value suggestion. We add it in revised manuscript. Comment 40: Line 56: Rephrase: “Therefore _ alternative phylloxera surveillance methods, models to predict phylloxera occurrence and strict quarantine measures might prevent the spread of the pest into regions or countries where grape phylloxera has not yet been recorded.” Response to comment 40: Thanks for your value suggestion. We correct it in revised manuscript. Comment 41: Line 56: Do you mean the “climate change” or in general “climatic changes” Response to comment 41: Thanks for your value suggestion. We change it in revised manuscript. Comment 42: Line 56: Rephrase: through changes” to e.g. “by affecting” Response to comment 42: Thanks for your value suggestion. We change it according reviewer’s suggestion in revised manuscript. Comment 43: Line 57: Check: “growth rates” Response to comment 43: Thanks for your value suggestion. We change it in revised manuscript. Comment 44: Line 57: Specify: “reproductive pressure” and “transmission” Response to comment 44: Thanks for your value suggestion. We added the references for these two points in revised manuscript. Comment 45: Lines 57/58: Rephrase: “significantly affecting their _distribution.” Response to comment 45: Thanks for your value suggestion. We change it in revised manuscript. Comment 46: Line 60: Rephrase: “expansion of grape phylloxera” Response to comment 46: Thanks for your value suggestion. We change it in revised manuscript. Comment 47: Line 62: Omit: “some” Response to comment 47: Thanks for your value suggestion. We delete it in revised manuscript. Comment 48: Line 63: Omit: “have” Response to comment 48: Thanks for your value suggestion. We delete it in revised manuscript. Comment 48: Line 70: Check: “global” twice the word Response to comment 48: Thanks for your careful work. We change it in revised manuscript. Comment 49: Line: 73: Change: “policies for the management and control of pest.” to “policies for pest management and control.” Response to comment 49: Thanks for your careful work. We change it in revised manuscript. Comment 50: Lines 131-135: Irrelevant information for the M&M sections. Could be moved to discussion or taken out. Response to comment 50: Thanks for your value suggestion. We delete this sentence in revised manuscript. Comment 51: Line 162: “value” different writing style Response to comment 51: Thanks for your careful work. We change it in revised manuscript. Comment 52: Lines 187-202: The suitability of grape phylloxera in most of the named regions is primarily explained by the presence of viticulture providing host plants and observations/reports due to the interest of phylloxera monitoring. If previously published phylloxera field observations (Supplement) serve as data basis for the presented study, they bias the conducted analysis. Therefore, for me, it is not correct to link the geographical distribution areas of D. vitifoliae in current viticultural areas with the analyzed climate data only, as done here (current invasive patterns). However, I would agree to apply your results/models to pyhlloxera distribution in potentially novel distribution areas without a current viticulture (lines 196-198, future invasive patterns). I suggest to be very clear in this point and maybe combine and rewrite parts of the two paragraphs. As mentioned in the previous round of comments, for linking climate factors and phylloxera distributions within traditional viticultural regions, a homogeneous host distribution in the analyzed areas/subregions is needed. Response to comment 52: Thanks for your value suggestion. We added this content in revised manuscript. Comment 53: Lines 190-193: Please check this part. It is not clear to me. What do you mean by highly suitable? How is this defined? Response to comment 53: Thanks for your value suggestion, we add the statement for the classification of suitable range it in revised manuscript. Comment 54: Lines 200-201: What do you mean with “ … suitable habitat area […] was projected to be 1.94×108 km2.”? In particular it is not clear how you correlate area units (square km) to habitat suitability. Therefore for me Table 3 (including the legend) is not self explaining. Response to comment 54: Thanks for your value suggestion. We added some statement for this content in revised manuscript. The environmental variables were choice with 2.5 arc min (~ 5km) spatial resolution. In fact, the potential distribution is a series of grids. The number of grids is calculated by the ArcGIS and multiplied by the area of each grid to get the area of the potential distribution area. Comment 55: Figure 1: Figure 1 is first mentioned in the Material and Methods section. Therefore it should be moved there or be mentioned in the Results section. Check that all other figures are placed after the paragraph they are named firstly through the manuscript. Response to comment 55: Thanks for your value suggestion and we moved it in the Result section. Comment 56: Line 200: km2 should be correct to km2 or km-2 according to the journal unit requirements. Response to comment 56: Thanks for your careful work, we correct it in revised manuscript. Comment 57: Table 3: The first column (pest) might not be necessary. Response to comment 57: Thanks for your value suggestion, we delete this column in revised manuscript. Comment 58: Figure 3: Is not self explaining. Axes units and the parameters are missing are missing. Response to comment 58: Thanks for your value work. We add the explanation in the legend. Comment 59: Lines 230/232: Twice “However,” Response to comment 59: Thanks for your careful work, we changed it in revised manuscript. Comment 60: Line 336-337: Climate factors are undoubtly important for the distribution of insects (pest) species. However, in case of specialist species (such as D. vitifoliae) the presence of a matching Vitis spp. host is essential in first place. I suggest to differentiate between generalist and specialist or milder the statement here. Response to comment 60: Thanks for your value suggestion. We changed the statement and added the content for generalist and specialist in revised manuscript. Comment 61: Line 343: Add: “The current study focuses on leaf-feeding grape phylloxera populations and did not consider root-feeding populations.” Response to comment 61: Thanks for your value suggestion. We add this in revised manuscript. Comment 62: Lines 343-345: Omit, because it is not done yet:” However, study suggest that root-feeding form cause most economic damage [12]. Thus, these 343 factors, including soil humidity and temperature, grape phylloxera form, especially for host-plant 344 availability [14] were considered in future study.” You might add a sentence that your results/factors might be interesting/groundbreaking for future studies investigating the distribution of root-feeding phylloxera instead. Response to comment 62: Thanks for your value suggestion. We changed it in revised manuscript. Comment 63: Line 326: Omit: “These studies all shown good simulation results.” Response to comment 63: Thanks for your value work. We deleted it in revised manuscript. Comment 64: Line 330-332: Check and rephrase more precisely: “However, many biotic factors, such as interspecific interactions [49] (e.g. inter- or intra species competition and 331 propagule pressure) also affect the potential range of species [49].” Response to comment 64: Thanks for your value suggestion, we delete this sentence in revised manuscript. Comment 65: Line 333: Omit: “also” Response to comment 65: Thanks for your value suggestion. We deleted it in revised manuscript. Comment 66: Line 333: Add: “grapevine hosts (Vitis spp.) Response to comment 66: Thanks for your value suggestion. We added it in revised manuscript. Comment 67: Line 334: Omit one comma Response to comment 67: Thanks for your careful work. We delete it in revised manuscript. Comment 68: Line 335: Unformatted citation Response to comment 68: Thanks for your careful work. We changed it in revised manuscript. Comment 69: Line 341: The use of leaf galling phylloxera only must should be mentioned from the beginning (e.g. Abstract and Material and Methods), to be clear. Response to comment 69: Thanks for your value suggestion, we added this information in Abstract section. Comment 70: Line 434: Add: “The current study focuses on leaf-feeding grape phylloxera populations and did not consider root-feeding populations.” Response to comment 70: Thanks for your value suggestion, we add it in revised manuscript. Comment 71: Line 340: Modify: “In additional, beside the drought stress and insect 340 pests, plant pathogens, also impact the physiology and health on vine [53].” to “Besides abiotic stresses such as drought, plant pathogens and/or other insect pests impact the host physiology with significant impact on D. vitifoliae populations [53].” Response to comment 71: Thanks for your value suggestion, we change it in revised manuscript.